# Salivary microbiome profiles for different clinical phenotypes of pituitary adenomas by single-molecular long-read sequencing

Xuefei Ji,[1] Pingping Li,[1] Qinglong Guo,[1] Liao Guan,[1] Peng Gao,[1] Bingshan Wu,[1] Hongwei Cheng,[1] Jin Xiao,[1] Lei Ye[1]

**ABSTRACT**  Pituitary adenomas (PAs) are common benign brain tumors. Although associations between the gastrointestinal microbiome and PA have been reported previously, studies on the distributions of salivary microbial species in PA patients and among their different clinical phenotypes are currently lacking. In this study, saliva samples from 42 patients and 20 healthy individuals were selected for third-generation sequencing. The PA group included four clinical phenotypes: adrenocorticotropic hormone-secreting PA ($n = 6$), growth hormone-secreting PA ($n = 9$), prolactin-secreting PA ($n = 18$), and nonfunctioning PA ($n = 9$). All samples were sequenced, and the data were clustered and de-chimerized to obtain information regarding the abundance of operational taxonomic units. We found that the species distributions in the saliva of PA patients were more abundant than those of healthy individuals. A total of 82 genera were identified across all samples, of which 14 and 17 genera were more abundant in the saliva samples of patients with PA and healthy individuals, respectively. In the phenotypic functional prediction, the phenotypes of anaerobic and Gram-positive organisms were more commonly seen in patients with PA than in healthy individuals. The bioinformatics prediction indicated that multiple metabolic pathways were involved in the pathogenesis of PA. In conclusion, this study highlighted the associations of salivary microbiome profiles with PA, which may improve the existing understanding of the pathogenesis of PA and provide diagnostic and therapeutic targets for PA.

**IMPORTANCE**  The gut and salivary microbiomes have been widely reported to be significantly associated with a number of neurological disorders. The stability of the microbiome in the oral cavity makes it a potentially ideal sample that can be conveniently obtained for the investigation of microbiome-based pathogenesis in diseases. In the present study, we used a single-molecule long-read sequencing technique to study the distribution of the salivary microbiota in patients with pituitary adenoma (PA) and healthy individuals, as well as among four clinical phenotypes of PA. We found that the diversity of salivary microbes was more abundant in PA patients than in healthy individuals. We also observed some unique genera in different PA phenotypes. The bioinformatics-based functional predictions identified potential links between microbes and different clinical phenotypes of PA. This study improves the existing understanding of the pathogenesis of PA and may provide diagnostic and therapeutic targets for PA.

**KEYWORDS**  pituitary adenomas, saliva microbiome, clinical phenotype, diagnostics

Pituitary adenoma (PA) is a common central nervous system (CNS) tumor that usually causes changes in endocrine hormone levels or induces optic nerve compression (1). On the basis of hormone abnormalities and histochemical evidence, PA is clinically classified as growth hormone-secreting PA (GH-PA), adrenocorticotropic hormone-secreting PA (ACTH-PA), prolactin-secreting PA (PRL-PA), and nonfunctioning PA (NF-PA) (2–4). Although medication treatment can be applied to some subgroups of PA, such

Address correspondence to Lei Ye, yelei@ahmu.edu.cn, Jin Xiao, xjtulip@163.com, or Hongwei Cheng, hongwei.cheng@ahmu.edu.cn.

Xuefei Ji, Pingping Li, and Qinglong Guo contributed equally to this article. The order of the authors was determined by the degree of contribution of each author in this study.

The authors declare no conflict of interest.

See the funding table on p. 10.

as bromocriptine for PRL-PA treatment, neurosurgical intervention remains the optimal therapeutic option that can relieve both optical and endocrine-related symptoms. However, because the pathogenesis of PA is not fully elucidated, effective therapeutic methods for controlling its development are currently unavailable.

The gut microbiome has recently gained considerable research interest because of its characteristics as a community of microorganisms that changes with an individual's growth and eventually reaches a stable state in adults (5, 6). An increasing number of investigations have reported associations between the gut microbiome and multisystem diseases such as gastrointestinal disease (7), metabolic disease (8), systemic inflammatory disease (9), neuropsychiatric disease (10), and multiple cancers (11). The novel theory of the "brain-gut axis" has been proposed, and some studies have reported that the gut microbiome has a significant influence on the pathogenesis of neurological diseases such as Alzheimer's disease (12), Parkinson's disease (13), intracranial aneurysms (14), and gliomas (15).

The distribution of the gut microbiome encompasses the entire alimentary tract, from the oral cavity to the rectum. Although the microbiome distributions in different regions of the digestive tract have been previously reported to be relatively independent, the microbiome in the upper gastrointestinal tract has been suggested to influence the distribution in the lower gastrointestinal tract (16). The microbiome in the oral cavity shows the lowest β-diversity and the highest α-diversity in comparison with other regions of the gastrointestinal tract (17, 18). Since healthy individuals show only slight variations in the composition of the microbiome in the oral cavity and saliva samples can be easily obtained and stored, the distribution of the oral microbiome may serve as a promising biomarker for diseases. In this regard, the association between the oral microbiome and neurological disorders has also been demonstrated. Sun et al. found that the oral salivary microbiomes in patients with high-risk ischemic stroke (IS) and IS show a higher diversity, and that the differential bacteria show some predictive value for the severity and prognosis of IS (19). Liu et al. observed that the relative abundance of *Moraxella*, *Leptotrichia*, and *Sphaerochaeta* in the saliva of patients with Alzheimer's disease greatly increases, whereas that of *Rothia* is significantly reduced (20). Additionally, differences in the abundances of the salivary microbiome have been proven to be associated with other neurological disorders such as Parkinson's disease (21) and glioma (22). We found three PA-related studies on the association between the gut microbiome and PA. Lin et al. investigated the correlations between the gut microbiome and GH-PA by performing metagenomics sequencing and found that patients with GH-PA show a reduced diversity of gut microbiota and increased levels of *Oscillibacter* and *Enterobacter* in comparison with healthy individuals (23). Hu et al. demonstrated dysbiosis of the gut microbiota between patients with invasive/noninvasive PA and healthy individuals (24). Furthermore, Nie et al. found that fecal microbiota transplantation of the intestinal flora of GH-PA patients promotes the growth of tumors in mouse models and increases soluble programmed death ligand-1 (sPD-L1) levels in peripheral blood samples (25). However, studies on the salivary microbiome profiles associated with different clinical phenotypes of PA are still unavailable.

In this study, using a single-molecular long-read sequencing method with saliva samples, we investigated the distribution of the oral microbiome between patients with PA and healthy individuals as well as across different clinical phenotypes of PA. We also intended to study their associations with the potential diagnosis and pathogenesis of different clinical phenotypes of PA. The scheme of the study is presented in Fig. 1.

## RESULTS

### Alpha- and beta-diversity analyses

We used α- and β-diversity analyses to evaluate the quality of the samples. We evaluated the observed species [Sob, reflecting the operational taxonomic unit (OTU) numbers], Chao (reflecting the species richness), Good's coverage (reflecting the coverage of OTUs with low abundance), Simpson (reflecting the species diversity), Shannon (reflecting the

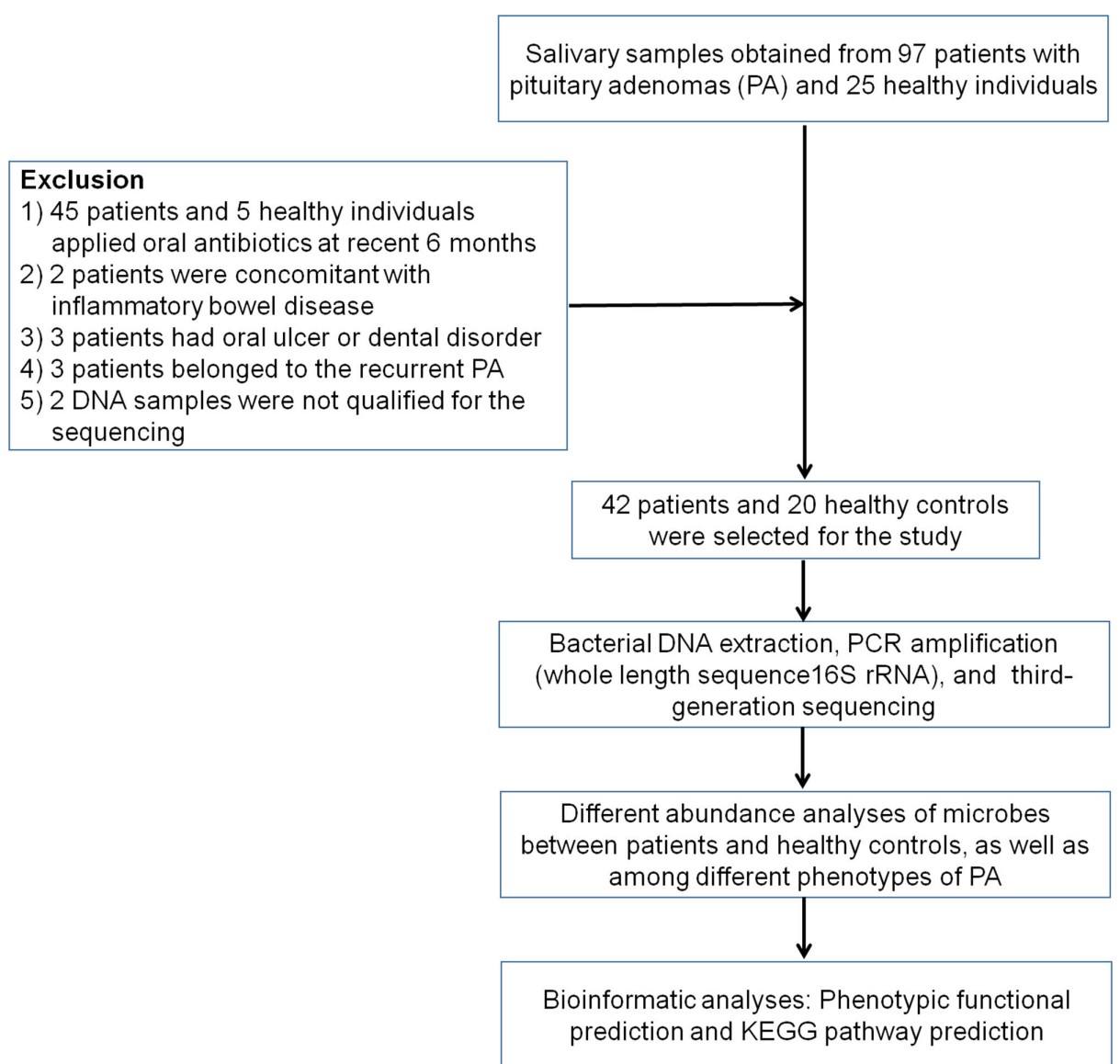

**FIG 1** The flow diagram of the experiment. KEGG, Kyoto Encyclopedia of Genes and Genomes.

species evenness), and phylogenetic diversity (PD)-whole tree (reflecting the coverage of OTUs with low abundance) indices in the α-diversity analysis. The comparison between patients with PA and healthy individuals is summarized in Fig. S1. Generally, the species distributions in the saliva of patients with PA were more abundant than those of healthy individuals. Additionally, α-diversity analysis was performed in the case-only cohort (Fig. S2). The results showed that the diversities in the GH- and NF-PA patients were generally higher than those in the ACTH- and PRL-PA patients. Principal coordinate analysis for β-diversity assessments in all samples (Fig. S3A) and the case-only samples (Fig. S3B) showed significant differences in sample characteristics between the patients with PA and healthy individuals. However, the samples among the four PA groups showed no significant differences. Because these four groups represented different tumor categories, the indistinguishable results among these four groups also indicated that the samples in the different groups were comparable.

## Distribution of microbes in patients with PA and healthy individuals

First, we analyzed the genus-level distribution in the salivary microbiome of patients with PA and healthy individuals. Microbes belonging to a total of 82 genera were found in all samples. Among these, 10 and 11 genera were more abundant in the salivary samples of patients with PA and healthy individuals, respectively (Fig. 2A). The results indicated that the top 10 most abundant genera in patients with PA and healthy individuals were *Streptococcus*, *Prevotella*, *Rothia*, *Porphyromonas*, *Neisseria*, *Granulicatella*, *Veillonella*, *Gemella*, *Haemophilus*, and *Peptostreptococcus* (Fig. 2B). We drew a Circos plot to depict the correlation between these top 10 microbial genera and the samples in different groups (Fig. 2C). We also analyzed the distributions of microbes between patients with PA and healthy individuals (Fig. 2D). The results showed that *Rothia*, *Porphyromonas*, *Peptostreptococcus*, *Oribacterium*, *Schaalia*, *Lancefieldella*, *Actinomyces*, *Tannerella*, *Corynebacterium*, *Mogibacterium*, *Arachnia*, *Butyrivibrio*, *Cardiobacterium*, and *Shuttleworthia* were abundant in the saliva of patients with PA, while *Neisseria*, *Veillonella*, *Haemophilus*, *Fusobacterium*, *Prevotellamassilia*, *Campylobacter*, *Alloprevotella*, *Megasphaera*, *Aggregatibacter*, *Bergeyella*, *Dialister*, *Centipeda*, *Shigella*, *Bacteroides*, *Casaltella*, *Simonsiella*, and *Klebsiella* were more abundant in the saliva of healthy individuals.

## Distributions of microbes in different clinical phenotypes of PA

We also analyzed the genus-level distribution in the salivary microbiomes of patients showing different clinical phenotypes of PA. After performing 16S rRNA whole-length sequencing, we illustrated the distributions and proportions of microbes using a heatmap and histogram (Fig. 3A and B). The findings indicated that *Streptococcus*, *Rothia*, *Porphyromonas*, *Prevotella*, *Granulicatella*, *Gemella*, *Neisseria*, *Peptostreptococcus*, *Leptotrichia*, and *Veillonella* were the 10 most abundant genera among the four PA groups. We drew a Circos plot to depict the correlations of the top five genera with the samples in different groups (Fig. 3C). A total of 67 genera were identified among the four clinical phenotypes of PA. In comparisons of the microbial species showing different levels of abundance in different groups, 55 genera were found to be commonly abundant in all groups. One genus was uniquely abundant in the ACTH-PA group (*Staphylococcus*); five were uniquely abundant in the GH-PA group (*Bifidobacterium*, *Johnsonella*, *Pseudoramibacter*, *Slackia*, and *Sneathia*); three were uniquely abundant in the NF-PA group (*Desulfobulbus*, *Ligilactobacillus*, and *Phocaeicola*); and three were uniquely abundant in the PRL-PA group (*Eikenella*, *Eubacterium*, and *Fructilactobacillus*) (Fig. 3D; Table S1).

## Different abundances of microbes among the four PA groups

In this section, we investigated the microbial species biomarkers in the different groups. First, we compared the microbes in the four groups, and the top 10 abundant microbes with different sizes were depicted using a bubble diagram (Fig. 3E). Then, in order to identify the unique biomarkers in the respective groups, we set the NF group as the reference. We performed a random forest analysis to evaluate the contribution degree of different species between groups. The mean reductions in the accuracy and Gini values reflected the importance of species via accuracy and heterogeneity, respectively. The results are shown in Fig. S4A through F. We also performed a receiver operating curve analysis to investigate the capacity of the species to separate the different groups on the basis of the results of random forest analyses. *Lachnoanaerobaculum* [Area Under the Curve (AUC) = 0.91], *Capnocytophaga* (AUC = 0.93), and *Catonella* (AUC = 0.92) served as the most distinguishable biomarkers in the ACTH group (Fig. S4A and B), GH group (Fig. S4C and D), and PRL group (Fig. S4E and F), respectively, in comparison with the NF group.

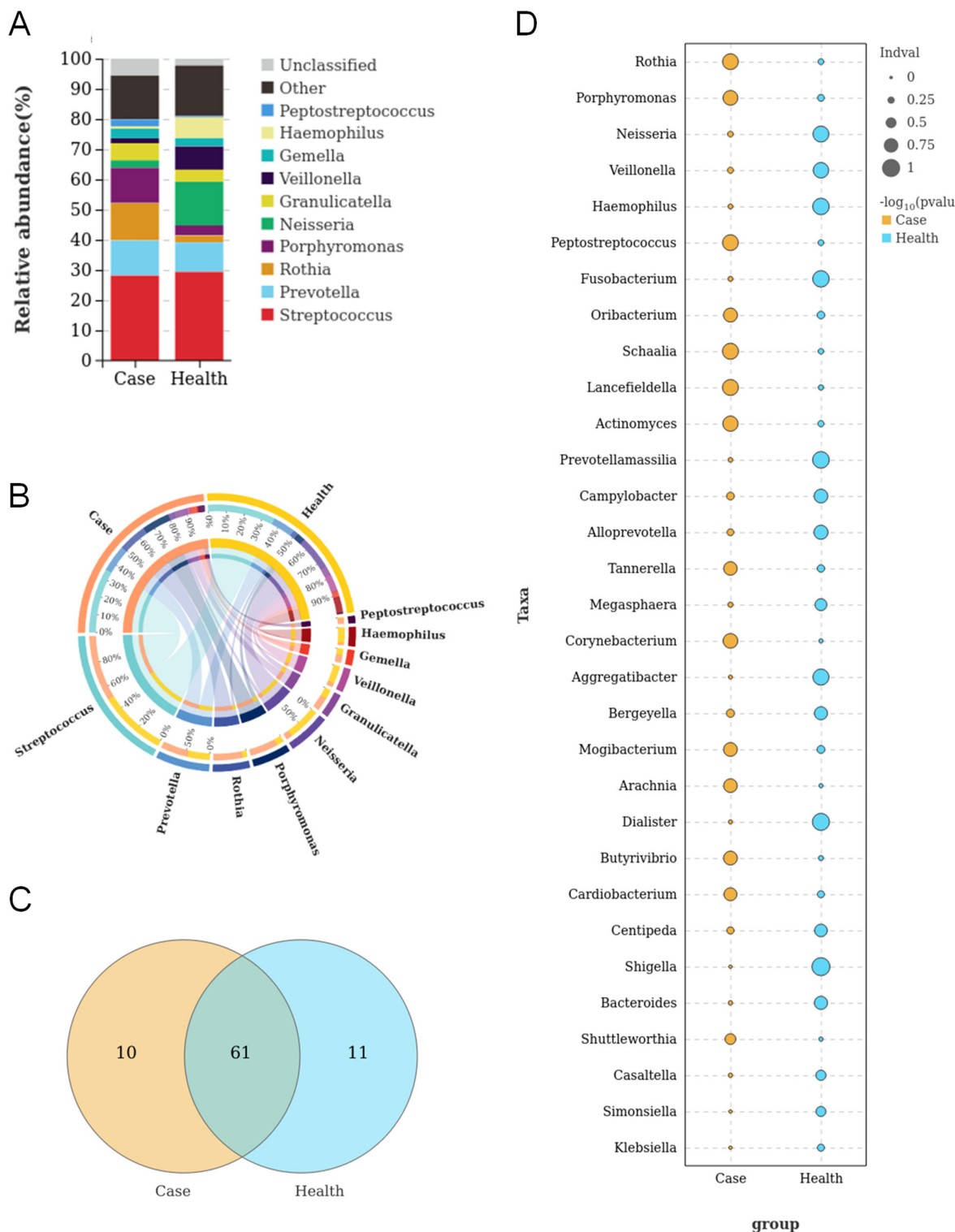

**FIG 2** Distribution of salivary microbiome in terms of the genus phylotype in samples between PA patients and healthy individuals. (A) Top 10 abundant microbes in the groups of PA patients and healthy individuals. (B) Circos plot indicating the interactions between the top 10 microbes and different groups. (C) Venn plot indicating the common and unique microbes among the two groups. (D) Differential abundance of microbes between the two groups.

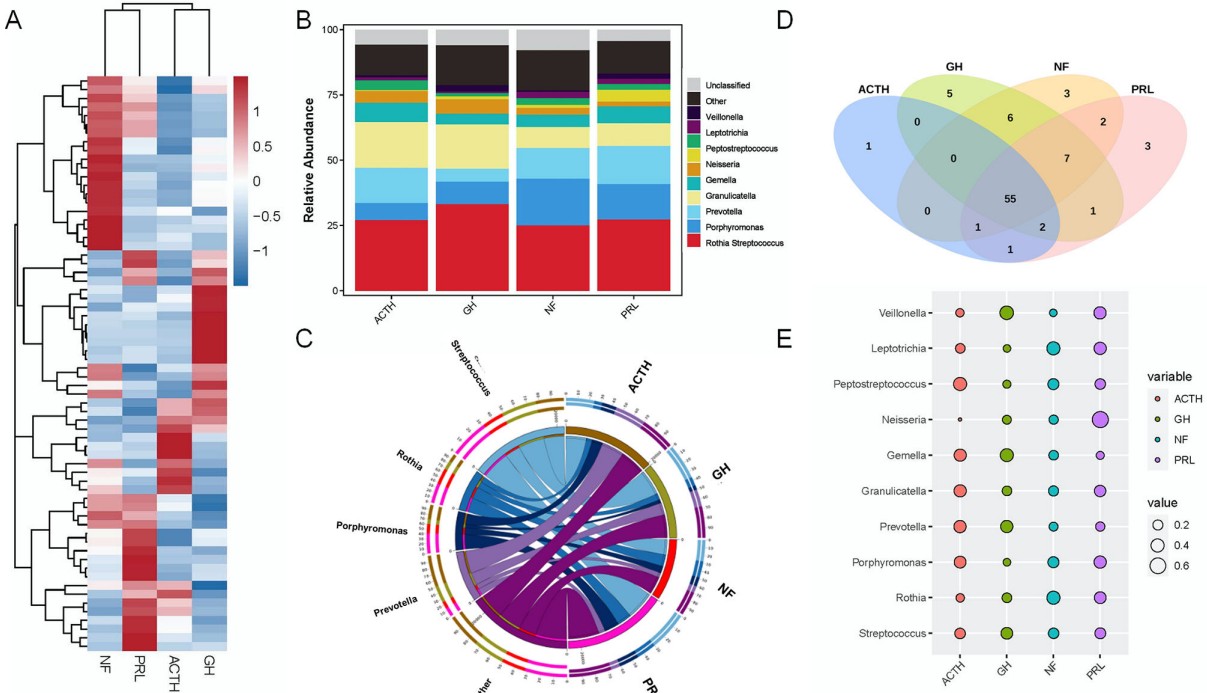

**FIG 3** Distribution of salivary microbiome in terms of the genus phylotype in samples from different groups of PA. (A) Heatmap for the distribution of the microbiome. The lines in the *y*-axis (left) indicating the clustering of microbes. (B) Proportions of different microbes in diverse groups. (C) Circos plot indicating the interactions between the top five microbes and different groups. (D) Venn plot indicating the common and unique microbes among the different groups. (E) Indicator analysis of the top 10 biomarkers among the different groups.

## Phenotypic functional prediction analysis

We used Bugbase software to perform phenotype predictions for the microbial species. First, we compared the phenotypes of microbial species between patients with PA and healthy individuals. The results indicated significant differences in the aerobic ($P < 0.001$), anaerobic ($P = 0.013$), mobile element-containing ($P < 0.001$), facultatively anaerobic ($P = 0.003$), Gram-negative ($P < 0.001$), Gram-positive ($P < 0.001$), pathogenic ($P < 0.001$), and oxidative stress-tolerant ($P < 0.001$) phenotypes. Among these, the anaerobic and Gram-positive phenotypes were more commonly seen in PA patients than in healthy individuals (Fig. 4A). Next, these phenotypes of microbial species were compared across PA subtypes. However, the results were all negative (Fig. S5). Finally, we used PICRUSt2 software to annotate the Kyoto Encyclopedia of Genes and Genomes (KEGG) pathway between patients with PA and healthy individuals. The top five annotated functions that may play important roles in PA pathogenesis were metabolism of cofactors and vitamins, carbohydrate metabolism, amino acid metabolism, metabolism of terpenoids and polyketides, and metabolism of other amino acids (Fig. 4A). The results indicated that the salivary microbes in PA patients might have higher levels of metabolism than those in healthy individuals.

## DISCUSSION

In this study, we explored the distribution of the salivary microbiome in healthy individuals and patients with four clinical phenotypes of PA. We found that the microbial diversities in patients with PA were higher than those in healthy individuals. Some salivary microbes were differentially abundant between these two groups. Furthermore, in the phenotypic functional prediction, the phenotypes of anaerobic and Gram-positive microbes were more commonly seen in patients with PA than in healthy individuals, while the aerobic, mobile element-containing, facultatively anaerobic, Gram-negative,

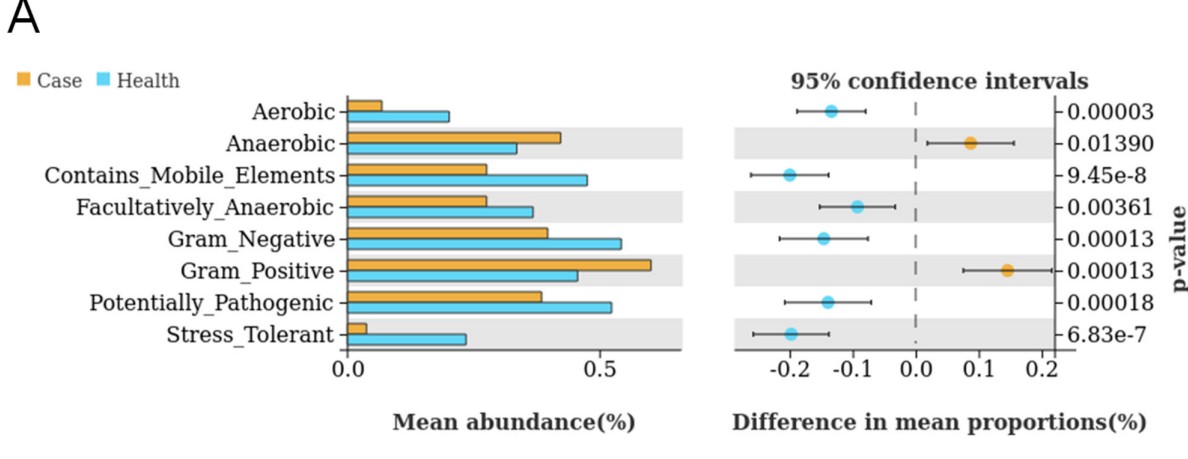

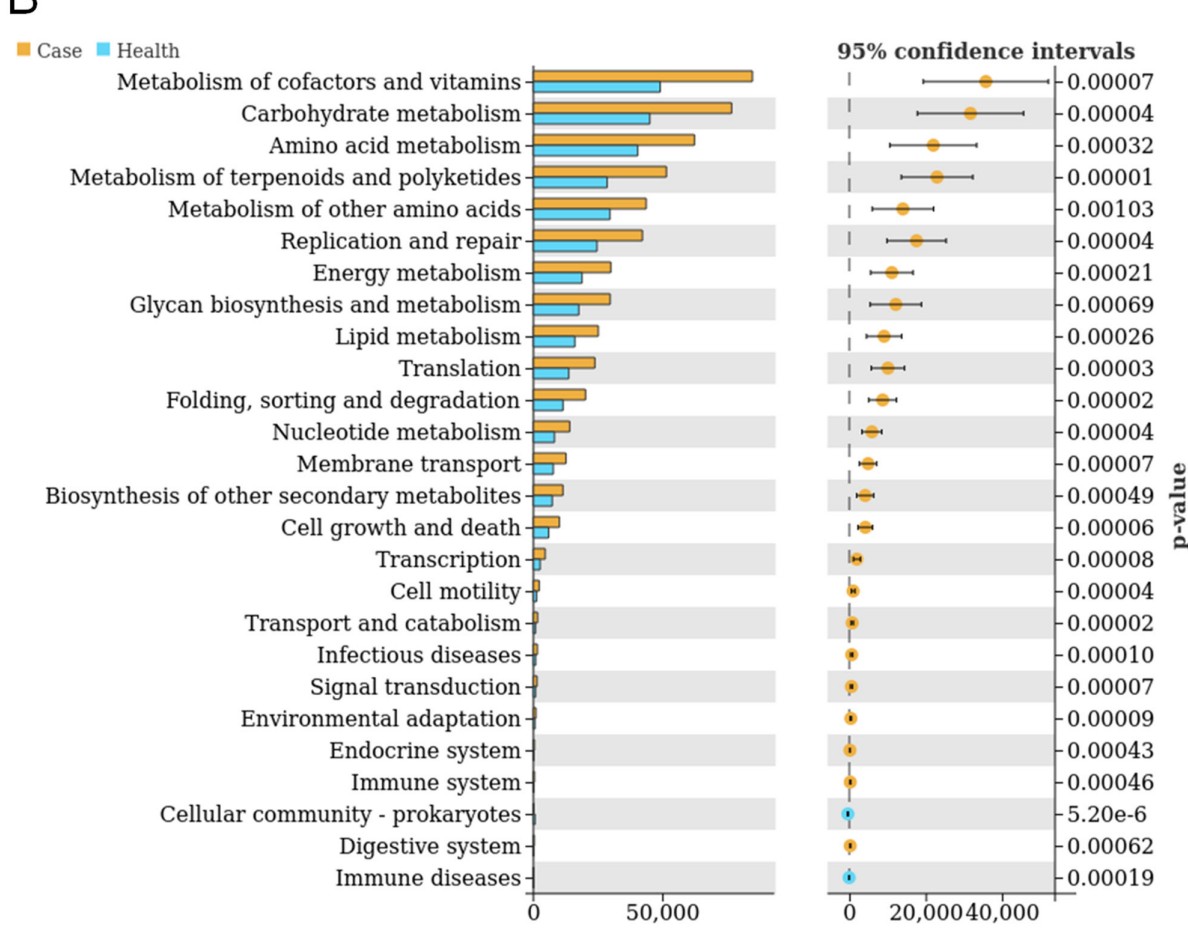

**FIG 4** Bioinformatics analyses for functional prediction. (A) Phenotype prediction for the microbial species between PA patients and healthy individuals. (B) KEGG pathway enrichment in the four groups.

pathogenic, and oxidative stress-tolerant phenotypes were significant in the healthy individuals. These results provided information about the association between salivary microbiome distributions and PA subtypes.

Previous studies have provided evidence that the salivary microbiome could significantly influence the distribution of the gastrointestinal microbiome and thereby

influence neurological disorders. Inflammation in the oral cavity exacerbates the intestinal inflammatory status by supplying colitis-causing bacteria and pathogenic T cells to the intestine (26) and potentially causing alterations of the gastrointestinal microbiome, which in turn affects the human neuroendocrine system through intestinal stimulation of the hypothalamic-pituitary-adrenal axis (27). Therefore, we inferred that the oral microbiome indirectly affected the neuroendocrine system through this mechanism. The oral microbiome may also have a direct association with neurological diseases. Narengaowa et al. found that periodontitis breaks down the oral mucosa, and that some toxins and inflammatory factors released from the oral microbiome bypass the damaged oral mucosa into the brain, causing inflammatory and immune responses in the central nervous system (28). Meanwhile, Liu et al. (29) also demonstrated that periodontitis causes a systemic inflammatory response that stimulates microglia in the CNS and mediates neuroinflammatory reactions. In a PA-related gut microbiome study, Lin et al. (23) found that the relative abundances of *Oscillibacter* and *Enterobacter* are remarkably higher in 10 patients with GH-PA than in 12 healthy individuals. However, these two genera were undetectable across all phenotypes of PA in our study (Table S1). This may be attributable to the differences in the composition of the microbiome between the oral cavity and the gastrointestinal tract. Additionally, we recently performed a case-only analysis of intratumoral microbes in different phenotypes of PA (30). However, we did not find an overlap between the salivary microbes and intratumoral microbes. Notably, these two studies used different sequencing platforms, and their intratumoral analyses did not include normal pituitary tissues, which are difficult to obtain. Therefore, the detailed influence of salivary microbes in the pathogenesis of PA requires further investigation. This study also provided a potential treatment target for different phenotypes of PA. With advancements in research on the microbiome, oral administration of microbes has also been proposed to have therapeutic potential (31). Weersma and Zhernakova (32) found that oral administration or fecal transplantation of specific bacteria promotes the immune response of the body, thereby improving the efficacy of antitumor therapy. However, whether this therapeutic approach has a beneficial effect on hormone-secreting pituitary tumors by affecting the neuroendocrine system requires extensive experimental evidence and additional research in animal studies.

The technique used in this study, third-generation sequencing (TGS), is also a topic of interest. Traditional studies for the detection of the microbiome have widely employed next-generation sequencing. In this approach, variable 1 to 9 areas in the microbiome's 16S rRNA are usually separately amplified by polymerase chain reaction (PCR), and splicing is performed to facilitate further sequencing. However, the transcriptomes that are formed by different splicing patterns are prone to producing multiple chimera errors, leading to barely perceptible misleading results. TGS technology uses a single-molecule real-time (SMRT) sequencing method in which the read length can reach 15 kb. Therefore, full-length 16S rRNA amplification achieves a better transcriptome of high quality, facilitating the investigation of RNA structural alterations such as alternative splicing, fusion genes, and allelic expressions (33, 34). Furthermore, TGS can detect more members of the microbiome at the genus or even species level.

Our study also had some notable limitations. Because of the strict qualifying conditions for patient inclusion, a relatively small sample size was finally selected for the experiment. Therefore, future studies should be conducted with an augmented sample size to increase the confidence of the results. However, the results of our experiments could not be verified with any certainty, and the mechanisms by which the microbiota change in response to disease have not yet been fully investigated. We suspect that the relationship between disease and microorganisms may be clearly explained when all the microorganisms are identified, without exceptions.

In conclusion, in this analysis of the association between the salivary microbiome and PA, we found that the diversity of salivary microbes is more abundant in PA patients

than in healthy individuals. The findings also provided predictive information about the potential microbe-based mechanisms in the pathogenesis of PA.

## MATERIALS AND METHODS

### Sample collection

A total of 97 patients, all of whom were diagnosed with PAs by two senior doctors according to the 2017 World Health Organization pituitary adenoma classification, and 25 healthy individuals were recruited from the Department of Neurosurgery, the First Affiliated Hospital of Anhui Medical University, between January 2019 and June 2021 (4). Among the recruited patients, 6 were not willing to participate in the study; 45 had received oral antibiotic treatment in the past 6 mo; 2 had concomitant inflammatory bowel disease; 3 had oral ulcers or dental disorders; 3 showed recurrent PA; and salivary DNA samples from 2 patients did not qualify for the final tests. Thus, salivary samples from 42 patients with PA and 20 healthy individuals were selected for the final tests. On the basis of the endocrinology and pathological assessments, 6, 9, 18, and 9 patients were found to show the ACTH, GH, PRL, and NF phenotypes. Saliva samples from patients and healthy individuals were collected preoperatively into sterile vials and stored at $-80°C$. All of the recruited patients denied any history of malignancy or systemic inflammatory diseases. The study was conducted in accordance with the Declaration of Helsinki and approved by the Institutional Ethics Committee of the First Affiliated Hospital of Anhui Medical University (approval number: 20200068, approval date: March 2020). Informed consent was obtained from all participants.

### DNA extraction and PCR amplification

The main TGS instrument used in this assay was the Pacific Biosciences Sequel platform, which employs SMRT and Circular Consensus Sequencing sequencing modes. Microbial DNA was extracted using HiPure Soil DNA Kits (or HiPure Stool DNA Kits) (Magen, Guangzhou, China) according to the manufacturer's protocols. The 8-ng DNA templates were amplified with the whole-length sequence of 16S rRNA by PCR. The primer sequences were as follows: forward: GCATCCACTCGACTCTCGCGTAGRGTTYGA-TYMTGGCTCAG and

 reverse: GCATCAGAGACTGCGACGAGARGYTACCTTGTTACGACTT. After 25 PCR cycles, the target bands of the amplification products that were tested using 0.8% gel electrophoresis showed a molecular weight of ~1.6 kb.

### PacBio sequencing

Amplicons were evaluated with 2% agarose gels and purified using the AxyPrep DNA Gel Extraction Kit (Axygen Biosciences, Union City, CA, USA) according to the manufacturer's instructions. Sequencing libraries were generated using the SMRTbell TM Template Prep Kit (PacBio, Menlo Park, CA, USA) in accordance with the manufacturer's recommendations. The libraries were sequenced on the PacBio Sequel platform. The experiment of sequencing was performed by Genedenovo, Ltd., Co. (Guangzhou, China). The sequence data reported in this study were archived in the Sequence Read Archive with the accession number PRJNA1004034. The reliability of the data analysis can be affected by low-quality, primer splice sequences, etc. Therefore, a series of data quality control processes are carried out on the raw data coming off the machine to ensure the reliability of the analysis results.

### Taxonomy annotation

Taxonomic classifications were conducted using Basic Local Alignment Search Tool (BLAST) (version 2.6.0), which searched for representative OTU or amplicon sequence variant (ASV) sequences against the NCBI 16S ribosomal RNA Database (Bacteria and

Archaea) ([http://www.ncbi.nlm.nih.gov](http://www.ncbi.nlm.nih.gov)) (version 202101) using the best hit with strict criteria ($E$-value $<10e^{-5}$, query coverage ≥60%, and the following identity thresholds: sequence identity ≥92% indicated the same species; sequence identity ≥88% indicated the same genus; sequence identity ≥85% indicated the same family; sequence identity ≥80% indicated the same order; sequence identity ≥75% indicated the class; and sequence identity ≥70% indicated the phylum). If no BLAST hit was retained, the sequence was labeled as unclassified.

## ACKNOWLEDGMENTS

X.J. designed the experiment, collected the samples, made a brief discussion and drafted the manuscript. P.L. performed the experiments and the majority of bioinformatic analyses, as well as made a discussion. At the peer review processes, we supplemented some data for healthy controls. Q.G. took charge of all supplemented experiments and analyses, so we added him as the third co-first author.

## AUTHOR AFFILIATION

[1]Department of Neurosurgery, The First Affiliated Hospital of Anhui Medical University, Hefei, Anhui, China

## AUTHOR ORCIDs

Lei Ye [http://orcid.org/0000-0002-5079-8730](http://orcid.org/0000-0002-5079-8730)

## FUNDING

| Funder | Grant(s) | Author(s) |
| --- | --- | --- |
| National Natural Science Foundation of China (NSFC) | 81901238 | Lei Ye |
| University Natural Science Research Project of Anhui Province | KJ2019A0248 | Lei Ye |
| Co-construction Project of Anhui Medical University and Affiliated Hospital | 2021lcxk017 | Hongwei Cheng |

## AUTHOR CONTRIBUTIONS

Xuefei Ji, Conceptualization, Data curation, Formal analysis, Writing – original draft | Pingping Li, Conceptualization, Investigation, Methodology, Writing – original draft | Qinglong Guo, Formal analysis, Investigation | Liao Guan, Methodology, Resources | Peng Gao, Validation, Visualization | Bingshan Wu, Resources, Software | Hongwei Cheng, Conceptualization, Funding acquisition, Supervision, Writing – review and editing | Jin Xiao, Conceptualization, Project administration, Supervision, Writing – review and editing | Lei Ye, Conceptualization, Data curation, Funding acquisition, Methodology, Writing – review and editing

## ADDITIONAL FILES

The following material is available online.

## Supplemental Material

**Supplemental Figure S1 (Spectrum00234-23-S0001.tif).** Supplementary Figure S1. α-Diversity analysis of the observed species (A), Chao1 index (B), Good's coverage (C), Simpson index (D), Shannon index (E), and PD-whole tree index (F) between the groups of PA patients and healthy individuals.
**Supplemental Figure S2 (Spectrum00234-23-S0002.tif).** Supplemental Figure S2. α-Diversity analysis of the observed species (A), Chao1 index (B), Good's coverage (C),

Simpson index (D), Shannon index (E), and PD-whole tree index (F) for the different groups of samples.

**Supplemental Figure S3 (Spectrum00234-23-S0003.tif).** Supplemental Figure S3. β-Diversity analysis based on principal coordinate analysis (PCoA) of the samples. (A) PCoA between the groups of PA patients and healthy individuals. (B) PCoA among the four groups of PA patients.

**Supplemental Figure S4 (Spectrum00234-23-S0004.tif).** Supplemental Figure S4. Indicator salivary microbes corresponding to the genus phylotype among the different groups. (A) Random-forest analysis of the biomarkers between ACTH-PA and NF-PA. (B) Receiver operating curve (ROC) analysis of the biomarkers between ACTH-PA and NF-PA. (C) Random-forest analysis of the biomarkers between GH-PA and NF-PA. (D) ROC analysis of the biomarkers between GH-PA and NF-PA. (E) Random-forest analysis of the biomarkers between PRL-PA and NF-PA. (F) ROC analysis of the biomarkers between PRL-PA and NF-PA.

**Supplemental Figure S5 (spectrum00234-23-S0005.tif).** Supplemental Figure 5. Phenotype prediction for the microbial species among the different groups of PA. (A) Gram-positive; (B) Gram-negative; (C) biofilm-forming; (D) pathogenic; (E) mobile element-containing; (F) aerobic; (G) anaerobic; (H) facultatively anaerobic; (I) oxidative stress-tolerant.

**Supplemental legends (Spectrum00234-23-S0006.docx).** Legends of supplemental figures and table.

**Supplemental table (Spectrum00234-23-S0007.docx).** Supplemental Table S1: The list of common and unique microbes among different groups of patients with pituitary adenoma.

## Open Peer Review

**PEER REVIEW HISTORY (review-history.pdf).** An accounting of the reviewer comments and feedback.

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
