## [Reviewer comments · Microbiology Spectrum]

Microbiology Spectrum

Salivary microbiome profiles for different clinical phenotypes of pituitary adenomas by single-molecular long-read sequencing

Xuefei Ji, Pingping Li, Qinglong Guo, Liao Guan, Peng Gao, Bingshan Wu, Hongwei Cheng, Jin Xiao, and Lei Ye

Corresponding Author(s): Lei Ye, First Affiliated Hospital of Anhui Medical University

Review Timeline:

Submission Date:	January 14, 2023
Editorial Decision:	May 26, 2023
Revision Received:	August 16, 2023
Accepted:	August 21, 2023

Editor: Justin Kaspar

Reviewer(s): The reviewers have opted to remain anonymous.

Transaction Report:

DOI: <https://doi.org/10.1128/spectrum.00234-23>

May 26, 2023

Dr. Lei Ye
First Affiliated Hospital of Anhui Medical University
Jixi Road 218
Hefei
China

Re: Spectrum00234-23 (Saliva microbiome profiles for different clinical phenotypes of pituitary adenomas by single-molecular long-read sequencing)

Dear Dr. Lei Ye:

Two expert reviewers have provided comments regarding the aforementioned manuscript. While generally positive, several areas of the manuscript will need improvement prior to publication. These include i) providing SRA accession number for deposited raw reads (required), ii) overall grammar review for the manuscript, iii) enhancing data description within provided tables/figures/figure legends, and iv) providing better discussion on the significance of the data the the author's conclusions/findings. Additionally it is recommended to include language on how potential biases not accounted for in the original inclusion criteria may modify the study's results.

Link Not Available

Sincerely,

Justin Kaspar

Journals Department
Reviewer comments:

Reviewer #1 (Public repository details (Required)):

The study includes large microbiome datasets collected from saliva samples from 42 individuals with different types of pituitary adenoma.

Reviewer #1 (Comments for the Author):

Line 71: "indicative indicator species" sounds repetitive. Can the authors revise and improve this sentence?

Lines 72-73: This sentence needs to be revised. The word "the" before "links" needs to be removed.

Line 102: "We found" should be changed to "We found that".

Line 106: "observe" should be "observed".

Line 125: "clinical" should be "clinically".

Overall, a comprehensive and extensive grammar English review needs to be done in this manuscript.

Line 169: Is "Enterbacter" correct? Shouldn't it be "Enterobacter"?

Supplementary Table 1 is not presented as a table. The authors need to have their data in a table format.

Fig 2A: The data represented in the heatmap do not have the legends in the Y-axis (on the left of the image) or on the top of the image. The reader understands that the red color indicates an increase, and the blue color indicates a decrease, but what is increasing and/or decreasing? The authors need to add more information to this image.

Fig 3C: The words inside the graph are barely readable if you increase magnification in the image. Please revise these figures. Increase the words, if you can. If you can't, add the information in the figure legends so the readers are able to read.

Figure 4: The authors explanation of the results is descriptive. I recommend that the authors add their perspective, and make the correlation between the metabolic pathways found and the PA type. The authors should also include the significance of these interesting findings.

The study provides a lot of interesting information. I recommend that the authors include one more figure to summarize their findings.

Reviewer #2 (Comments for the Author):

Authors using a 42 patients cohort with pituitary adenoma to reveal the difference in saliva microbiome between PA subtypes. Subsequently, some specific microbes were identified as a biomarker in different PAs. Previous researches have uncovered the association between oral microbiomes and neurological disorders. Thus this work from authors might also provide a standpoint of oral microbiome in PAs. However, few more questions should be addressed in the following sections.

Major concern 1: The authors aim to figure out the inner connection between microbiome in saliva and several kinds of pituitary adenomas, which definitely provided a brand new sight in the pathology of PAs. However, tedious paragraph about gut microbiome rather than saliva was exhibited in the introduction section. In order to emphasize the great impact and promising diagnostic role of saliva, I strongly insist that the authors should provide fundamental information about saliva with neurological diseases and neuroendocrine tumors rather than repeat previous gut researches in PAs.

Major concern 2: As the authors noted in this article, saliva is a biospecimen with ideal accessibility. Therefore authors should better expand the sample size and clarify the inclusion criteria to remove the risk of bias, like the effect of diet, the usage of other medicine like antibiotics, with or without metabolic disorders like diabetes, etc.

Major concern 3: Here in this article, authors depict a new oral-intestinal-cerebral axis. Unfortunately, authors just described the difference in saliva microbiome varying from several PA subtypes. No solid evidence was provided to prove the existence of this axis.

Major concern 4: It is recommended to involve several healthy individuals. Then re-analyse the 16s sequencing data with all PAs. In this way can the authors provide more information related to the underlying mechanism that some specific microbiome in saliva may take a part in the development of PAs regardless of any subtypes.

Major concern 5: It is recommended that the authors can use mass spectrum to further identify the proteome among these subtypes and with healthy individuals. Thus the authors can discuss further on the protein levels rather than an ambiguous hypothesis.

Line 64: Deleta be.
Line 250: Change this into these.
Line 314: Delete a dot.
Line 336: Please provide the ethical approval number here.

Staff Comments:

Preparing Revision Guidelines

Please return the manuscript within 60 days; if you cannot complete the modification within this time period, please contact me. If you do not wish to modify the manuscript and prefer to submit it to another journal, please notify me of your decision immediately so that the manuscript may be formally withdrawn from consideration by Microbiology Spectrum.

Reviewer #1:

Comment 1: Line 71: "indicative indicator species" sounds repetitive. Can the authors revise and improve this sentence?

Answer: Thanks. We deleted the word "indicator".

Comment 2: Lines 72-73: This sentence needs to be revised. The word "the" before "links" needs to be removed.

Answer: Thanks. Corrected.

Comment 3: Line 102: "We found" should be changed to "We found that".

Answer: Thanks. Corrected.

Comment 4: Line 106: "observe" should be "observed".

Answer: Thanks. Corrected.

Comment 5: Line 125: "clinical" should be "clinically".

Answer: Thanks. Corrected.

Comment 6: Overall, a comprehensive and extensive grammar English review needs to be done in this manuscript.

Answer: We greatly appreciate the reviewer's kind suggestion. We noticed the language problems, so we polished the revised version of the manuscript with a commercial company to minimize any expression errors.

Comment 7: Line 169: Is "Enterbacter" correct? Shouldn't it be "Enterobacter"?

Answer: Thanks. Corrected.

Comment 8: Supplementary Table 1 is not presented as a table. The authors need to have their data in a table format.

Answer: We thank for your suggestive comment. We corrected the xls. file into a Table format.

Comment 9: Fig 2A: The data represented in the heatmap do not have the legends in the Y-axis (on the left of the image) or on the top of the image. The reader understands that the red color indicates an increase, and the blue color indicates a decrease, but what is increasing and/or decreasing? The authors need to add more information to this image.

Answer: We are grateful for your careful comment. We tried to present the legends in the Y-axis of Figure 3A (Figure 2A in the original version). Unfortunately, because there were many categories of microbes in genus phelotype that were identified in the experiment, it would be very tough to read the legends, even though Figure 3 takes up a whole layout of the printing page. The heatmap provides the profile information of identified microbes with color changes and clustering with lines (on the left). Based on these factors, we did not supplement legends of Y-axis. Those microbes with differential abundance were depicted in Figure 3B and 3E. Meanwhile, we supplemented an explanation for the lines in the figure legend.

Comment 10: Fig 3C: The words inside the graph are barely readable if you increase magnification in the image. Please revise these figures. Increase the words, if you can. If you can't, add the information in the figure legends so the readers are able to read.

Answer: Thanks very much for your advise. We agree the problem. Because we performed an extra experiment about the salivary microbiome of healthy individuals, we re-organized the composing of figure 3B-3E into Supplementary Figure 4. The size the figure was augmented and the words in the figure was readable.

Comment 11: Figure 4: The authors explanation of the results is descriptive. I recommend that the authors add their perspective, and make the correlation between the metabolic pathways found and the PA type. The authors should also include the significance of these interesting findings.

Answer: We thank very much for you kind suggestion. In the modified version of manuscript, we added 20 samples of healthy subjects and performed some analyses. Finally, we deleted the KEGG annotation among the subgroups of PA, but listed the KEGG annotation between the PA patients and healthy

subjects, which was potentially associated with the pathogenic mechanisms of PA. The results showed that most of annotated mechanisms were correlated to biological metabolisms, so we inferred that the salivary microbes in PA patients might had higher levels of metabolisms than that in the healthy subjects. However, because the KEGG annotation belongs to a predictive method, and we did not perform validation experiments, so we did not expand the section in the discussion section. It was supplemented at line 231-237.

Comment 12: The study provides a lot of interesting information. I recommend that the authors include one more figure to summarize their findings.

Answer: We greatly appreciate the reviewer's kind comment. Because our study focused on the clinical associations of salivary microbes with PA patients. We did not perform fundamental analysis for the microbes with pathogenesis of PA. Therefore, we think it is improper to present a hypothesis-based mechanism figure in the manuscript. After a consideration, we modified the flow diagram of the experiment (Figure 1). We think that figure summarizes the whole experiment of our study. Thank again for your kind suggestions.

Reviewer #2:

Authors using a 42 patients cohort with pituitary adenoma to reveal the difference in saliva microbiome between PA subtypes. Subsequently, some specific microbes were identified as a biomarker in different PAs. Previous researches have uncovered the association between oral microbiomes and neurological disorders. Thus this work from authors might also provide a standpoint of oral microbiome in PAs. However, few more questions should be addressed in the following sections.

Comment 1: The authors aim to figure out the inner connection between microbiome in saliva and several kinds of pituitary adenomas, which definitely provided a brand new sight in the pathology of PAs. However, tedious paragraph about gut microbiome rather than saliva was exhibited in the introduction section. In order to emphasize the great impact and promising

diagnostic role of saliva, I strongly insist that the authors should provide fundamental information about saliva with neurological diseases and neuroendocrine tumors rather than repeat previous gut researches in PAs.

Answer: Many thanks for your suggestive comment. We noticed that there might be some tedious in literatures concerning the gut microbiome in the introduction section. Therefore, we deleted the corresponding paragraphs and supplemented some contexts about the associations between the salivary microbiome and neurological disorders (line 110-118). Meanwhile, in this study, we aimed to study the associations of salivary microbiome between PA patients and healthy individuals, as well as among 4 clinical phenotypes of PA. However, although there are some reports about the establishments of PA modeling animal, such as using USP8^{-/-} mice for ATCH adenoma (Endokrynol Pol. 2023;74:181-189), and using oestrogen for prolactinoma (Endocr Relat Cancer. 2022;29:703-716), none of these methods were proved to be universally representative for PA modeling. Furthermore, there are still lack of modeling methods for some clinical phenotypes of PA, such as NF-PA. Therefore, we are unable to perform the mechanism studies using animal models to provide fundamental evidences between the salivary microbe with pathogenesis of PA. We would make some attempts to study the mechanism of certain microbe in the pathogenesis of PA in the near future.

Comment 2: As the authors noted in this article, saliva is a biospecimen with ideal accessibility. Therefore authors should better expand the sample size and clarify the inclusion criteria to remove the risk of bias, like the effect of diet, the usage of other medicine like antibiotics, with or without metabolic disorders like diabetes, etc.

Answer: We greatly appreciate your kind comments. As showed in Figure 1, we initially collected 97 PA patients. Meanwhile, we had a restrict inclusion criteria to minimize the bias. Therefore, among all recruited PA patients, 45 patients applied oral antibiotics at recent 6 month; 2 patients were concomitant with inflammatory bowel disease; 3 patients had oral ulcer or dental disorders; and 4 patients belonged to the recurrent PA. Additionally, none of PA patients and healthy subjects had special diets, such as ketogenic diet or Mediterranean diet, and suffered from metabolic disorders.

Also, we tried to supplement some samples of PA patients. However, we noticed that almost all people in China has experienced COVID-19 at December 2022 and June 2023, respectively, but the initial samples were all free of COVID-19. We can not eliminate the bias which was induced by the pandemics. Therefore, we did not supplement additional samples of PA patients in the revised version of manuscript. Meanwhile, 19 samples from healthy individuals were collected before December 2022 and were free of COVID-19 infection, so we supplemented the sequencing for microbiome of saliva samples in healthy subjects and analyzed the distribution differences between PA patients and healthy subjects.

Comment 3: Here in this article, authors depict a new oral-intestinal-cerebral axis. Unfortunately, authors just described the difference in saliva microbiome varying from several PA subtypes. No solid evidence was provided to prove the existence of this axis.

Answer: We thank very much for your comment. We agree. We make some literatures discussing the oral-intestinal-cerebral axis. However, we just performed the clinical association analyses of salivary microbiome between PA patients and healthy individuals, as well as among different clinical phenotypes of PA. we did not perform a fundamental study because the unavailability of PA modeling animals which were proved to be universally representative in laboratory studies. So we minimized the concept of “oral-intestinal-cerebral axis” in the context to avoid the hypothesis-based discussion.

Comment 4: It is recommended to involve several healthy individuals. Then re-analyze the 16s sequencing data with all PAs. In this way can the authors provide more information related to the underlying mechanism that some specific microbiome in saliva may take a part in the development of PAs regardless of any subtypes.

Answer: We are grateful for your suggestive comment. We collected 20 salivary samples from healthy subjects. Meanwhile, these individuals had the same exclusive criteria as that in PA patients. We performed whole-length

sequencing for 16S rRNA and made a re-analyze the distribution difference of species between PA patients and healthy subjects.

Quality controls of samples, including α - and β -diversity analyses, were showed at Supplementary Figure 1 and Supplementary Figure 3. There results of differentially abundant microbes between PA patient and healthy individuals were depicted at Figure 2, Figure 4, SFigure 1, and SFigure 3A. In the main context, we also supplemented the expressions and discussions on it (line 146-151, 154-156, 161-181, 221-237, 242-249).

Comment 5: It is recommended that the authors can use mass spectrum to further identify the proteome among these subtypes and with healthy individuals. Thus the authors can discuss further on the protine levels rather than an ambiguous hypothesis.

Answer: Thanks for your kind suggestion.

First, we should also honestly admitted that, all the saliva samples were performed with 16S rRNA sequencing, and the left amount of saliva is not sufficient for proteomic analysis. Additionally, it was suggested by a researcher to perform a multi-omics analysis in the study, such as an interactive analysis between microbiome-proteome. However, we realized that numerous proteins can not enter in to central nervous system, although pituitary tissue is at the outside of blood-brain barrier. Thus, if we perform a proteomic analysis, we can not distinguish the protein which is practically functioned in PA and may lead to misinformation to other researchers. In this study, we performed an association analysis of microbiome of saliva in PA patients and healthy subjects. We will reduce the part of hypothesis and increase more result-based contents.

Comment 6: Line 64: Deleta be.

Answer: Thanks. Corrected.

Comment 7: Line 250: Change this into these.

Answer: Thanks. Corrected.

Comment 8: Line 314: Delete a dot.

Answer: Thanks. Corrected.

Comment 9: Line 336: Please provide the ethical approval number here.

Answer: Thanks for the comment. We supplemented the number of ethical approval number and the copy of supporting material can be also uploaded with requested.

The study was conducted in accordance with the Declaration of Helsinki and approved by the Institutional Ethics Committee of the First Affiliated Hospital of Anhui Medical University (Approval number: 20200068, approval date: 2020 March).

August 21, 2023

Dr. Lei Ye
First Affiliated Hospital of Anhui Medical University
Jixi Road 218
Hefei
China

Re: Spectrum00234-23R1 (Salivary microbiome profiles for different clinical phenotypes of pituitary adenomas by single-molecular long-read sequencing)

Dear Dr. Lei Ye:

Your manuscript has been accepted, and I am forwarding it to the ASM Journals Department for publication. You will be notified when your proofs are ready to be viewed.

Sincerely,

Justin Kaspar
Editor, Microbiology Spectrum
